# The implications of initial model drift for decadal climate predictability using EC-Earth

Andreas Sterl<sup>1</sup>

<sup>1</sup>Royal Netherlands Meteorological Institute (KNMI), P.O. Box 201, NL-3730 AE De Bilt, Netherlands *Correspondence to:* Andreas Sterl (sterl@knmi.nl)

**Abstract.** The large heat capacity of the ocean as compared to the atmosphere provides a memory in the climate system that might have the potential for skilful climate predictions a few years ahead. However, experiments so far have only found limited predictability after accounting for the deterministic forcing signal provided by increased greenhouse gas concentrations. One of the problems is the drift that occurs when the model moves away from the initial conditions towards its own climate. This drift

5 is often larger than the decadal signal to be predicted. In this paper we describe the drift occurring in the North Atlantic Ocean in the EC-Earth climate model and relate it to the lack of decadal predictability in that region. While this drift may be resolution dependent and disappear in higher resolution models, we identify a second reason for the low predictability. A subsurface heat content anomaly can only influence de atmosphere if (deep) convection couples it to the surface, but the occurrence of deep convection events is random and probably mainly determined by unpredictable atmospheric noise.

## 10 1 Introduction

The heat capacity of the ocean is much larger than that of the atmosphere. The energy needed to change the temperature of the upper 4 m or so of the ocean is enough to change the temperature of the whole atmosphere by the same amount. Therefore the ocean provides a memory that potentially makes multi-year climate forecasts possible. Imagine, for instance, that a sizeable volume of the North Atlantic Ocean is warmer than normal. Releasing the extra energy over the course of a few years would

15 systematically warm the overlying atmosphere, and the prevailing westerly winds would advect the warmer air into Europe. This could make possible a prediction of the kind "the next few years will be warmer than average" in western Europe.

Recent years have seen a large amount of research into *decadal prediction*, i.e., prediction for up to ten years ahead (see the review by Meehl et al., 2014). Typically, climate models are initialized with observed past ocean states, and their ability to reproduce observed climate anomalies is assessed.

20 Climate variability can be externally forced or produced internally. The most important external forcing arises from the increasing greenhouse gas (GHG) concentrations, and the prediction "the next years will be warmer than average" is nearly always correct. This deterministic signal has to be removed from the model output to assess the predictability of internal variability arising from the ocean memory. After subtraction of the GHG signal, Van Oldenborgh et al. (2012) and Karspeck et al. (2015) found no predictability signal except for in the Labrador Sea. That decadal predictability is low after accounting

5

for GHG forcing was also noted by other authors, e.g., Chikamoto et al. (2013) and Boer et al. (2013). Predictability is further reduced if the model is driven without known volcanic aerosol loadings (Van Oldenborgh et al., 2012).

In short, the predictability of internally generated variability appears to be low in current climate models. In this paper we try to shed some light on this apparent lack of decadal predictability by analysing the initial drift occurring in the ocean part the EC-Earth climate model (Hazeleger et al., 2012; Sterl et al., 2012) and investigating its effect on the atmosphere. We concentrate on the North Atlantic Ocean because this is the region where most models find some kind of predictability.

## 2 Experimental set-up

The decadal prediction experiments described in this paper are performed with EC-Earth v2.3 (Hazeleger et al., 2012; Sterl et al., 2012). EC-Earth is a climate model which uses NEMO (Madec, 2008) in the ORCA1 configuration as its ocean component.

- 10 This configuration has an average horizontal resolution of 1°, which, due to the curvilinear grid employed, increases to  $\approx 0.5^{\circ}$ in the Labrador Sea region. The initialization procedure described in Wouters et al. (2013) is used. Ocean initial conditions are taken from ECMWF's ORAS4 ocean reanalysis (Balmaseda et al., 2013), and atmosphere initial conditions are from ERA-Interim (Dee et al., 2011). ORAS4 comprises five members, and five atmosphere initial conditions were created from ERA-Interim using singular vector perturbation. Three of the ORAS4 members are used and combined with all five atmosphere
- 15 conditions, creating three ensembles differing by their ocean initial conditions. Hindcasts are started on the 1st of November of the years 1960, 1965, 1970, 1985, 1990, 1995, 2000, and 2005. Unfortunately, sea-ice restart files for 1975 and 1980 as used by Wouters et al. (2013) where not available any more. The forecast length is 10 years.

The original aim of this research was to investigate how initialization in different ocean regions impacts prediction quality. To test this two ensembles were performed in which the ocean restart files were downgraded in specific regions. The impact

20 turned out to be low, and results from these ensembles are presented here alongside those using the complete restart files. Essentially, the low impact of the degraded restarts was the reason to write the present paper.

The decadal prediction runs are compared to the ORAS4 ocean reanalysis (Balmaseda et al., 2013). The comparison concentrates on the North Atlantic Ocean, as earlier work (e.g., Van Oldenborgh et al.; 2012, Hazeleger et al., 2013a) has shown that this is one of the few regions in the world where useful skill exists beyond the trend caused by the greenhouse effect.

### 25 3 Model drift in decadal predictions

## 3.1 Initialization strategies

To perform a decadal climate prediction the climate model has to be initialized. The initialization of the atmosphere component is not crucial as the atmosphere time scale is only a few weeks. The atmosphere "forgets" its initial state quickly and cannot contribute to the ability of the model to forecast the climate years ahead, as is the purpose of decadal climate prediction. Any

model skill must come from the ocean with its large thermal inertia and correspondingly long time scales. Initialization of the

30

ocean component of the climate model is therefore crucial.

There are basically two methods to initialize the ocean, namely full-field and anomaly initialization. In full-field initialization an oceanic state is constructed from observations and used as the initial state. In anomaly initialization an observation-based anomaly field is added to the model climatology. In theory the full-field initialization should be preferable as the prediction run starts from a situation close to the real oceanic state. In practice, however, this "real" state is often incompatible with the

5 model's own climatology. As a result the model immediately starts to drift away from the initial state towards its own climate. The drift may be large (e.g., Hazeleger et al., 2013b; Huang et al., 2015; Sanches-Gomez et al., 2016) and can obscure the climate signal to be predicted. In practice it turns out that the skill of both initialization strategies is comparable (e.g., Polkova et al., 2014; Hazeleger et al., 2013b; Smith et al., 2013).

This paper aims at describing the initial drift and its implications for model performance and decadal predictions. To show
that this drift is indeed towards the "model's own climate", comparisons with results from the PD and PI runs of Sterl et al. (2012) will be made. Those runs were performed with constant present-day (PD) and pre-industrial (PI) forcing, using the same version of EC-Earth (v2.3).

## **3.2** Determining the drift

To isolate the climate signal from full-field initialized decadal prediction runs, the model drift has to be determined and 15 subtracted from the individual forecasts. Following the recommendation prepared for CMIP5 by the CMIP-WGCM-WGSIP Decadal Climate Prediction Panel (ICPO, 2011), the drift is assumed to be independent of the start date of the forecast, and only dependent on lead time (= time since start date). Thus the assumption is that for an ensemble of forecasts the development of any variable T can be written as

$$T(t_l, t_s, i) = T_{\text{drift}}(t_l) + T_{\text{signal}}(t_l, t_s) + T_{\text{noise}}(t_l, t_s, i),$$
(1)

20 where  $t_s$  denotes the start date,  $t_l$  the lead time, and *i* the ensemble member.  $T_{drift}$  is the common drift, depending on lead time, but not on start date,  $T_{signal}$  is the signal, which is the same for each member with the same start date, and  $T_{noise}$  is the residual.

Averaging (1) over all start dates and ensemble members gives an estimate of  $T_{\text{drift}}(t_l)$ ,

$$\hat{T}_{\rm drift}(t_l) = \frac{1}{n} \sum_{t_s \in T_s} \frac{1}{m} \sum_{i=1}^m T(t_l, t_s, i),$$
(2)

where *n* is the number of start dates,  $T_s = \{t_s\}$  is the set of all start dates, and *m* denotes the number of ensemble members. In our case n = 8,  $T_s = \{1960, 1965, 1970, 1985, 1990, 1995, 2000, 2005\}$ , and m = 5. In deriving (2) we have assumed that  $T_{\text{signal}}$  is uncorrelated between start dates, and that the noise is uncorrelated between start dates and between members of the same start date.

An estimate of the signal can now be obtained by subtracting the estimated drift and averaging over the ensemble members,

30 
$$\hat{T}_{\text{signal}}(t_l, t_s) = \frac{1}{m} \sum_{i=1}^m (T(t_l, t_s, i) - \hat{T}_{\text{drift}}(t_l)).$$
 (3)

5

20

Note that this procedure naturally removes the annual cycle, as the latter is contained in  $\hat{T}_{drift}(t_l)$ . Therefore, drift-corrected model output will be compared to anomalies from ORAS4, where anomalies are calculated in the usual way by subtracting the mean annual cycle.

It is considered good practise (ICPO, 2011) to apply (2) in a cross-validated manner by excluding the current start date from the calculation of the drift,

$$\hat{T}_{\text{drift}}(t_l, t_s) = \frac{1}{n-1} \sum_{t_{s'} \in T_s \setminus \{t_s\}} \frac{1}{m} \sum_{i=1}^m T(t_l, t_{s'}, i).$$
(4)

As this paper is mainly concerned with the drift itself rather than with the correction of the forecast, (2) is used throughout.

## 3.3 Overturning Stream Function and Subpolar Gyre strength

The drift in EC-Earth is large and affects the whole North Atlantic basin. We illustrate this by investigating the changes in the
Atlantic Meridional Overturning Circulation (AMOC) and the barotropic circulation in the basin. The AMOC is described by
the overturning stream function

$$\Phi(y,z;t) = \int_{x_W}^{x_E} dx \int_{z}^{0} dz' v(x,y,z';t),$$
(5)

where  $x_W$  and  $x_E$  are respectively the western and eastern boundaries of the basin, and the barotropic circulation by the barotropic stream function

15 
$$\Psi(x,y;t) = -\int_{x_W}^x dx' \int_{z_B}^0 dz \, v(x',y,z;t),$$
(6)

where  $z_B$  is the ocean depth.

The upper panel of Fig. 1 shows the time-evolution of the AMOC strength, smoothed by a 12 month running mean (12mrm) filter. The AMOC strength is defined as the maximum of  $\Phi$  between 20°N and 50°N (using other limits within the North Atlantic Ocean does not change the conclusions) and over the whole depth. The thick black curve represents the values from ORAS4, while the coloured curves are for single members of the experiments. Clearly, the evolution of the AMOC in the experiments is dominated by a common drift and has hardly any resemblance with the ORAS4-values. The middle panel

shows the drift-corrected curves (anomalies for ORAS4). Clearly the experiments do not contain any useful signal.

The drift itself is shown in the lower panel of Fig. 1. After an increase by more than 1 Sv during the first two years the AMOC strength declines by 5 Sv, from  $\approx 22$  Sv to  $\approx 17$  Sv. The latter value is close to the climatological value of  $\approx 16.5$  Sv as reported

by Sterl et al. (2012) for EC-Earth. The drift is indeed a "return to the model's own climatology". To put the magnitude of the drift into perspective it should be compared to the reduction of  $2.7 \pm 2.3$  Sv between 2004-2008 and 2008-2012 as observed by the RAPID/MOCHA array (Smeed et al., 2014), or the  $\approx 8$  Sv reduction between 1995 and 2003 than can be inferred from Fig. 1 for ORAS4. The initial model drift is large, but within observational constraints. On the other hand the model variability

5