# Peer review of "The implications of initial model drift for decadal climate predictability using EC-Earth"

_Ocean Science, 2016_

## Referee Comment (RC1) · Anonymous Referee #1 · 20 Jul 2016

This paper presents an analysis of decadal prediction experiments conducted with the EC-Earth climate model which focuses on the drift in these full-field-initialized ensemble simulations. I found much of the analysis to be interesting and novel because it is rare to see an explicit focus on the drift in initialized climate prediction studies. In the abstract, the author proposes to "describe" the drift and then "relate it to the lack of ... predictability" in the North Atlantic. The paper succeeds well enough in describing the drift, with a nice set of figures and adequate writing (although there are numerous instances where presentation clarity could be improved, as noted below). However, I don't think the manuscript actually succeeds in shedding much light on the low predictability seen in the EC-Earth prediction system.

Part of the problem may be a lack of specificity throughout the manuscript about what is meant by "the lack of decadal predictability" (abstract). The EC-Earth v2.3 system

analyzed herein exhibits non-trivial skill at predicting SPG heat content at decadal lead times (Hazeleger et al. 2013). This author focuses on the evidently low predictability of the large scale circulation and subpolar heat fluxes (although no quantitative skill scores are given). The analysis of drift (up through section 3.5) does not automatically inform the lack of predictability, and the logic used to draw conclusions in sections 4 and 5 seems flawed (see my specific reactions below). The authors fail to cite and discuss their results in the context of published studies in which significant decadal prediction skill is seen in the North Atlantic in full-field initialized ensembles despite the presence of large drift. By the end, it seems evident that the real focus of the paper is skill at predicting air-sea heat flux, which should probably be clearly stated up front.

The "second reason for the low predictability" (abstract) presented here (that the deep convection required to communicate ocean heat content signals to the atmosphere is inherently unpredictable) is rather hand-wavy and not convincingly supported by the analysis. Figure 6 is an analysis of intra-ensemble spread in heat (and salt) content. By itself, this doesn't shed much light on heat flux skill scores, which could depend more on (presumably large) inter-ensemble (across start dates) heat content differences (ie, if there were large heat content differences in the SPG from before the mid-1990s to after the mid-1990s, as observations clearly show, then even average (and/or poorly predicted) winter mixing should tap that heat content signal and be apparent to some degree in heat flux). Jumping straight to a conclusion about Figure 7 from Figure 6 (P10.L28-34) ignores this, and hence obfuscates the interpretation of how this decadal prediction system is working (or, isn't working).

The conclusion that "the drift in the atmosphere is not caused by the drift in the ocean" (P14.L13), may well be true, but it hasn't been demonstrated. This statement is apparently based on the different timescales in Figs. 7 and 9, but these represent very different regional averages – what exactly can be concluded by this comparison? Certainly nothing so strong as "an ocean signal that far exceeds the internal variability of the model is not able to impact the atmosphere." On the contrary, I'd be surprised if the

dramatic cooling in T2m off of the Grand Banks (Fig. 8, lower panel) were not related (indeed, driven by) the drift in the position of the model North Atlantic Current (Fig. 4). In short, strong and surprisingly general conclusions are drawn in this manuscript that do not really follow from what is shown.

A serious and major revision would be required, in my opinion, to transform this into a publication-worthy study with clear, strong, and adequately-supported conclusions.

Specific comments:

* Abstract: Awkward first sentence.

* P1.L8: "the" instead of "de"

* P1.L24: "Labrador Sea" seems too restrictive. Karspeck et al. (2015) found enhanced predictability associated with initialization in a broad subpolar gyre region, and Van Oldenborgh et al. (2012) report skill in the "northern North Atlantic."

* P2.L2: The analysis of CMIP3 models with and without volcanic aerosol forcing in Van Oldenborgh et al. (2012) *suggests* that initialized decadal predictions would be less skillful without foreknowledge of volcanoes, but doesn't actually demonstrate that.

* P2.L17: "were" instead of "where"

* Fig. 1: Why not extend the time axis to show the full plume of 2005-initialized ensemble? Thick colored lines (ensemble means) are almost impossible to see.

* P4.L15: This (unconventional) integration from the western boundary results in negative (positive) streamfunction in the subtropical (subpolar) gyres (see Fig. 3). Probably worth mentioning, if not changing the streamfunction sign in order to be consistent with common usage.

* P4.L23: The "increase by more than 1 Sv during the first two years" seen in Figure 1 hides a substantial initial drop of ~2 Sv, does it not? As noted in the caption, the 12mrm smoothing results in curves that do not start from ORAS4, but clearly there

must a sharp reduction in AMOC during the first 6 months of the predictions, followed by the increase. Can the authors comment on what the drift looks like without 12mrm-smoothing?

* P4.L28: What exactly is meant by "within observational constraints"?

* Fig. 2: Why are only two ensembles shown in bottom panel?

* P5.L9: To my eye, the SPG still flows into the Labrador Sea in PD.

* P5.L10: Define "GS"

* Fig. 3: Please specify the time intervals used for computing mean AMOC and BSF, for both ORAS4 and PD.

* P6.L1: Suggest "absent" instead of "lost"

* Fig. 4: I am confused by the averaging over the first two months for BSF (upper right panel). The rationale is that "the strength of the SPG declines rapidly in the first year." However, Fig. 2 (top panel) shows that SPG strength is always higher than ORAS4 at the start of the 12mrm plumes, indicating that SPG strength rapidly increases during the first 6 months. See also above related comment for AMOC at P4.L23 – why is annual mean OK for AMOC in Fig. 4 top panel given the rapid decline in AMOC strength in the first 6 months? Please clarify.

* P6.L9-P9.L7: This discussion regarding the importance of "maintaining the current structure in and around the LS", and the implication that eddy-resolving resolution is crucial for that, is too vague and speculative, in my opinion. It seems to me that maintaining convection in the LS is key to maintaining robust gyre circulation there, and apparently this model cannot maintain convection there (P9.L11). Plenty of other O(1-degree) coupled models are able to.

* P9.L9: "is" instead of "comprises of" ?

* P10.L8: "is" instead of "occurs" ?
* Fig. 6: I'm a bit confused about what exactly is being plotted here. Is SPG the region from Fig. 3? Does PCC show the pattern correlation over the SPG region at each time and depth? How is RMSD normalized? Is the "drift" substracted computed just from the 1995 ensemble(s) or is it the drift according to Eq. 2?

* P10.L20: I suggest coming up with distinct names for the boxes shown in Fig. 3 so that there is no confusion about what is meant by "SPG". As it stands, "SPG" means different things for different variables.

* P10.L21: I find it hard to see what's happening on seasonal timescales in Fig. 7.

* P10.L25: What is meant by "distinguished ... signal"?

* P10.L27: I'm confused. Presumably, Fig. 7 is showing net downward heat flux, with negative climatological values indicating that the atmosphere tends to cool the ocean in the SPG. Then, why would a drift towards more negative fluxes (Fig. 7 lower panel) indicate "less heat is extracted from the ocean"?

* P10.L29: explanation for the low predictability of what, exactly?

* Fig. 8: Why don't you actually show the difference from observed climatology from the PD simulation for T2m and SLP, to support the argument that the model drift represents a return to model's own climatology?

* Fig. 9: Anomalies from what climatology? I find this an unconvincing plot, because I see a lot of variation with time at all latitudes (and particularly in the 40-60N range). I do not see how this shows that "the drift takes place within the first year".

* P14.L8: Again, it's not clear to me that the drift represents a reduction in the amount of heating of the atmosphere (see comment on P10.L27 above)

* P14.L8-L20: I find the logic difficult to follow, and hence unconvincing, in this discussion. When I look at Fig. 8 (bottom panel), I see a large cooling signal in the vicinity of Grand Banks and throughout the SPG that is almost certainly related to the ocean

model drift (ie, the loss of a realistic North Atlantic Current pathway and the overall weakening of the overturning and gyre circulations that transport heat into the SPG). How exactly does the author come to the conclusion that "an ocean signal that far exceeds the internal variability of the model is not able to impact the atmosphere"?

* P14.L23: "casting doubt" instead "causing doubts"

* P14.L25: I don't think this conclusion is obvious at all, particularly since numerous other decadal prediction studies have demonstrated convincing skill in the North Atlantic (Robson et al. 2012, doi:10.1029/2012GL053370; Yeager et al. 2012, doi:10.1175/JCLI-D-11-00595.1; Hermanson et al. 2014, doi:10.1002/2014GL060420; Yeager et al. 2015, doi:10.1002/2015GL065364). Perhaps these results indeed cast doubts on the feasibility of using EC-Earth for predictions, but such a sweeping conclusion is wholly unjustified.

---

## Referee Comment (RC2) · Anonymous Referee #2 · 29 Jul 2016

This paper is on the drifts in the EC-Earth model hindcasts after full field initialisation with ERA-Interim atmosphere and ORAS4 ocean. The drifts are interesting to see although they are not presented in enough detail or with focussed enough diagnostics to really learn a lot about the causes of drift. Where these are discussed the paper appears very speculative and the results shown do not really justify the conclusions. I therefore think that the claim of the paper to have explained why decadal hindcasts appear to fail with this model (and by implication to the same deficiencies in other models) cannot really be justified. I cannot therefore recommend publication in the full journal The presentation is rather superficial level at times Page 3 L2 full field initialisation the ocean state is not simply "constructed from Observations" L6-7 the drift occurring in full-field assimilation is exactly the same as in seasonal forecasting approaches which have accepted forecasting skill If the drift is to obscure forecast skill the system must

become very non-linear L16 One might expect drift to at least have a seasonally de­pendent component (which is stated later top of page 4) Section 3.3 Using the AMOC to assess drift relative to an observation based reanalysis truth is rather hazardous I would say. There is not a lot of evidence that the AMOC is not really robustly re­producable in these reanalysis products yet (as Karspeck et al note) so using it as a forecast target is insecure. Page 5 The low natural variability in EC-Earth cannot be a surprise given this is such a low resolution model that has not really be tuned against low frequency variability signals. Indeed this whole section 3.3 seems to end with the conclusion that a higher resolution version of the model might be better for the AMOC but there is no evidence presented for this apart from noting another model has found this. Section 3.4 discussed vertical structure of drift in AMOC and Lab sea. The lack of predictability as well as the drift in the AMOC is then suggested to be related to the rapid decorrelation and drift of surface heat fluxes and upper ocean properties in the Lab sea in different model members. But the hypothesis is not really proved. In addition there is no discussion of the way the Lab sea / SPG water gets out and at what depths it propagates down to influence the AMOC. I would argue this is an aspect which will be greatly affected by model resolution. Section 3.5 is very brief and discusses the ob­vious result that the atmosphere is un predictable on short timescales. The discussion section 4 then presents some speculative Lagrangian argument for the pathways of air from a colder SPG towards Europe. But none of the diagnostics are lagrangian and no attempt is made at budgeting the movement of heat content anomalies. The section concludes that this argument does not agree with the model results anyway and that therefore "obviously the mechanism does not work"

---

## Editor Comment (EC1) · M. Hecht (Editor) · 8 Aug 2016

Dear Dr. Sterl, while the reviewers appreciate aspects of the manuscript, a very major revision will be required in order to overcome all or most of their objections. I will leave it to your good judgement to decide on what course to take. I realize that decadal prediction is a tough problem with a dearth of easy results (I once heard it characterized as the problem with the least in the way of "low hanging fruit").

Wishing you the best. Sincerely, –Matthew Hecht

———————————————————

---

## Author Comment (AC1) · 2 Sep 2016

Dear Reviewer,

Thank you very much for taking the time to thoroughly review my paper. Please find my answers to your comments in red below. My conclusion is that I do not have the time for the extra work needed to overcome the concerns raised by you and the reviewer. Therefore, and I will not try to submit a revised version of the paper.

With kind regards,

[Figure]

Andreas Sterl

This paper is on the drifts in the EC-Earth model hindcasts after full field initialisation with ERA-Interim atmosphere and ORAS4 ocean. The drifts are interesting to see although they are not presented in enough detail or with focussed enough diagnostics to really learn a lot about the causes of drift. Where these are discussed the paper appears very speculative and the results shown do not really justify the conclusions. I therefore think that the claim of the paper to have explained why decadal hindcasts appear to fail with this model (and by implication to the same deficiencies in other models) cannot really be justified. I cannot therefore recommend publication in the full journal The presentation is rather superficial level at times.

- Page 3 L2 full field initialisation the ocean state is not simply "constructed from Observations" The initial state is a reconstruction of the T and S fields that is based on observations, while in anomaly initialization only the anomalies are based on observations.

- L6-7 the drift occurring in full-field assimilation is exactly the same as in seasonal forecasting approaches which have accepted forecasting skill If the drift is to obscure forecast skill the system must become very non-linear After a few months the atmosphere is in equilibrium with the upper ocean and decoupled from the deep ocean. The latter can only be reached through intermittent deep convection. This is a highly non-linear process.

- L16 One might expect drift to at least have a seasonally dependent component (which is stated later top of page 4) I did not investigate whether the drift has a seasonal component, but it might be interesting to do so. - On top of page 4 I do not state that such a dependence exists. I just state that eq. (3) results in a $\hat{T}_{\text{signal}}$ that by construction does not contain an annual cycle, and thus must be

compared to anomalies. - More to seasonality: It might also be interesting to investigate the dependence of the drift on start month. All runs investigated in the paper were initialized on November, 1$^{st}$. Would the drift look differently for runs initialized on, say, May, 1$^{st}$?

- Section 3.3 Using the AMOC to assess drift relative to an observation based reanalysis truth is rather hazardous I would say. There is not a lot of evidence that the AMOC is not really robustly reproducable in these reanalysis products yet (as Karspeck et al note) so using it as a forecast target is insecure. Thank you for pointing this out. My main point in the discussion of the AMOC is, however, that its development is similar in all runs, irrespective of start data ("common drift"), and towards the model's own climatology.

- Page 5 The low natural variability in EC-Earth cannot be a surprise given this is such a low resolution model that has not really be tuned against low frequency variability signals. I do not say that I am surprised. Indeed this whole section 3.3 seems to end with the conclusion that a higher resolution version of the model might be better for the AMOC but there is no evidence presented for this apart from noting another model has found this. Indeed it would be better to have hi-res runs with the same model. Unfortunately, I had no access to any.

- Section 3.4 discussed vertical structure of drift in AMOC and Lab sea. The lack of predictability as well as the drift in the AMOC is then suggested to be related to the rapid decorrelation and drift of surface heat fluxes and upper ocean properties in the Lab sea in different model members. But the hypothesis is not really proved. I do not say that the drift is related to the rapid decorrelation of heat flux and upper ocean properties. It is the limited predictability that is. In addition there is no discussion of the way the Lab sea / SPG water gets out and at what depths it propagates down to influence the AMOC. I would argue this is an aspect which will be greatly affected by model resolution. Presumably, yes. As said above, I

had no access to such runs.

- Section 3.5 is very brief and discusses the obvious result that the atmosphere is un predictable on short timescales. Here I disagree. I show that the drift in the atmosphere occurs over a much shorter period than the drift in the ocean and thus that the atmosphere is not predictable on *long* time scales.

- The discussion section 4 then presents some speculative Lagrangian argument for the pathways of air from a colder SPG towards Europe. But none of the diagnostics are lagrangian and no attempt is made at budgeting the movement of heat content anomalies. The section concludes that this argument does not agree with the model results anyway and that therefore "obviously the mechanism does not work" I agree that a thorough Lagrangian heat budget analysis would improve the paper.

---

## Author Comment (AC2) · 2 Sep 2016

Dear Reviewer,

Thank you very much for taking the time to thoroughly review my paper. Please find my answers to your comments in red below. My conclusion is that I do not have the time for the extra work needed to overcome the concerns raised by you and the second reviewer. Therefore, and I will not try to submit a revised version of the paper.

With kind regards,

Andreas Sterl

This paper presents an analysis of decadal prediction experiments conducted with the EC-Earth climate model which focuses on the drift in these full-field-initialized ensemble simulations. I found much of the analysis to be interesting and novel because it is rare to see an explicit focus on the drift in initialized climate prediction studies. In the abstract, the author proposes to "describe" the drift and then "relate it to the lack of ... predictability" in the North Atlantic. The paper succeeds well enough in describing the drift, with a nice set of figures and adequate writing (although there are numerous instances where presentation clarity could be improved, as noted below). However, I don't think the manuscript actually succeeds in shedding much light on the low predictability seen in the EC-Earth prediction system.

Part of the problem may be a lack of specificity throughout the manuscript about what is meant by "the lack of decadal predictability" (abstract). The EC-Earth v2.3 system analyzed herein exhibits non-trivial skill at predicting SPG heat content at decadal lead times (Hazeleger et al. 2013). This author focuses on the evidently low predictability of the large scale circulation and subpolar heat fluxes (although no quantitative skill scores are given). I agree that giving quantitative skill scores would improve the paper. The analysis of drift (up through section 3.5) does not automatically inform the lack of predictability, and the logic used to draw conclusions in sections 4 and 5 seems flawed (see my specific reactions below). The authors fail to cite and discuss their results in the context of published studies in which significant decadal prediction skill is seen in the North Atlantic in full-field initialized ensembles despite the presence of large drift. It is right that drift *per se* does not preclude predictability. My point, however, is that in the model the large drift in the ocean has no correspondence in the atmosphere. While the atmosphere has an initial drift, its time scale is much shorter (two years or so; Figs. 7c and 9) than that of the drift in the ocean (nearly ten years; Fig. 1c). So the large ocean signal (drift) fails to impact the atmosphere. On the year-to-year time scale the heat

flux contains no systematic signal (Fig. 7a), even not after detrending (Fig. 7b). Thus also on this time scale there is no systematic impact of the ocean on the atmosphere. By the end, it seems evident that the real focus of the paper is skill at predicting air-sea heat flux, which should probably be clearly stated up front. Right.

The "second reason for the low predictability" (abstract) presented here (that the deep convection required to communicate ocean heat content signals to the atmosphere is inherently unpredictable) is rather hand-wavy and not convincingly supported by the analysis. I agree that the analysis could be improved. Figure 6 is an analysis of intra-ensemble spread in heat (and salt) content. By itself, this doesn't shed much light on heat flux skill scores, which could depend more on (presumably large) inter-ensemble (across start dates) heat content differences (ie, if there were large heat content differences in the SPG from before the mid-1990s to after the mid-1990s, as observations clearly show, then even average (and/or poorly predicted) winter mixing should tap that heat content signal and be apparent to some degree in heat flux). Jumping straight to a conclusion about Figure 7 from Figure 6 (P10.L28-34) ignores this, and hence obfuscates the interpretation of how this decadal prediction system is working (or, isn't working). You are right. Heat content in the SPG is indeed showing a marked increase after 1990, and the hindcasts are able to reproduce it. However, a corresponding signal is missing in the heat flux.

The conclusion that "the drift in the atmosphere is not caused by the drift in the ocean" (P14.L13), may well be true, but it hasn't been demonstrated. This statement is apparently based on the different timescales in Figs. 7 and 9, but these represent very different regional averages – what exactly can be concluded by this comparison? Certainly nothing so strong as "an ocean signal that far exceeds the internal variability of the model is not able to impact the atmosphere." On the contrary, I'd be surprised if the dramatic cooling in T2m off of the Grand Banks (Fig. 8, lower panel) were not related (indeed, driven by) the drift in the position of the model North Atlantic Current (Fig. 4). In short, strong and surprisingly general conclusions are drawn in this manuscript that do not really follow from what is shown. I agree that a more detailed analysis is

desirable.

A serious and major revision would be required, in my opinion, to transform this into a publication-worthy study with clear, strong, and adequately-supported conclusions.

**Specific comments:**

- Abstract: Awkward first sentence. OK.

- P1.L8: "the" instead of "de" Thanks.

- P1.L24: "Labrador Sea" seems too restrictive. Karspeck et al. (2015) found enhanced predictability associated with initialization in a broad subpolar gyre region, and Van Oldenborgh et al. (2012) report skill in the "northern North Atlantic." Yes, "Subpolar Gyre Region" would better describe the results in the cited papers.

- P2.L2: The analysis of CMIP3 models with and without volcanic aerosol forcing in Van Oldenborgh et al. (2012) *suggests* that initialized decadal predictions would be less skillful without foreknowledge of volcanoes, but doesn't actually demonstrate that. To my opinion their Fig. 6c+d is a demonstration.

- P2.L17: "were" instead of "where" Thanks.

- Fig. 1: Why not extend the time axis to show the full plume of 2005-initialized ensemble? For technical reasons the 2005-initialized runs end in 2009. This should have been mentioned. Thick colored lines (ensemble means) are almost impossible to see. Right. I should have left them out as they do not add extra information.

- P4.L15: This (unconventional) integration from the western boundary results in negative (positive) streamfunction in the subtropical (subpolar) gyres (see Fig. 3).

Probably worth mentioning, if not changing the streamfunction sign in order to be consistent with common usage. Right.

- P4.L23: The "increase by more than 1 Sv during the first two years" seen in Figure 1 hides a substantial initial drop of $\approx 2$ Sv, does it not? As noted in the caption, the 12mrm smoothing results in curves that do not start from ORAS4, but clearly there must a sharp reduction in AMOC during the first 6 months of the predictions, followed by the increase. Can the authors comment on what the drift looks like without 12mrm smoothing? A sharp drop occurs during the first one or two months.

- P4.L28: What exactly is meant by "within observational constraints"? It refers to the preceding sentence which presents evidence that a change of the AMOC strength of several Sv within a few years has been observed. The drift in the hindcast runs is inside the observation-based bandwidth of change.

- Fig. 2: Why are only two ensembles shown in bottom panel? I calculated SPG strength only for two ensembles. Also the upper panels of the figure contain data from only two ensembles (10 runs).

- P5.L9: To my eye, the SPG still flows into the Labrador Sea in PD. That is right. The sentence should read "[...] and the SPG only reaches the southern part of the LS."

- P5.L10: Define "GS" Right - Gulf Stream

- Fig. 3: Please specify the time intervals used for computing mean AMOC and BSF, for both ORAS4 and PD. ORAS4: 1958-2009, PD: model years 2400-2450 as indicated by the time axes in lower row.

- P6.L1: Suggest "absent" instead of "lost" OK

- Fig. 4: I am confused by the averaging over the first two months for BSF (upper right panel). The rationale is that "the strength of the SPG declines rapidly in the first year." However, Fig. 2 (top panel) shows that SPG strength is always higher than ORAS4 at the start of the 12mrm plumes, indicating that SPG strength rapidly increases during the first 6 months. See also above related comment for AMOC at P4.L23 – why is annual mean OK for AMOC in Fig. 4 top panel given the rapid decline in AMOC strength in the first 6 months? Please clarify. I apologize. An important information is missing in the caption of Fig. 2: The rapid drop in SPG strength during the first few months is so large that I had to add 20 Sv to the model results in Fig. 2a to have curves in the range spanned by ORAS4.

- P6.L9-P9.L7: This discussion regarding the importance of "maintaining the current structure in and around the LS", and the implication that eddy-resolving resolution is crucial for that, is too vague and speculative, in my opinion. It seems to me that maintaining convection in the LS is key to maintaining robust gyre circulation there, and apparently this model cannot maintain convection there (P9.L11). Plenty of other O(1- degree) coupled models are able to. The discussion can be improved upon, I agree. Especially, runs with a higher resolution would be helpful to make the point. Unfortunately, such runs are not yet available for me. - A more detailed analysis could also shed light on the hen-and-egg problem: does the gyre circulation change because convection in the LS ceases, or does convection cease because the gyre structure changes?

- P9.L9: "is" instead of "comprises of" ? OK

- P10.L8: "is" instead of "occurs" ? OK

- Fig. 6: I'm a bit confused about what exactly is being plotted here. Is SPG the region from Fig. 3? Yes. Does PCC show the pattern correlation over the SPG

region at each time and depth? Yes. How is RMSD normalized? By the spatial variances of the two time series:

$$\text{RMSD}(t) = < \frac{(a(x,y,t) - b(x,y,t))^2}{\sqrt{(\sigma_a^2(x,y) + \sigma_b^2(x,y))}} >, \qquad (1)$$

where $< ... >$ denotes spatial (over $x$ and $y$) averaging. Is the "drift" substracted computed just from the 1995 ensemble(s) or is it the drift according to Eq. 2? The latter, i.e., the average over all start dates and ensemble members.

- P10.L20: I suggest coming up with distinct names for the boxes shown in Fig. 3 so that there is no confusion about what is meant by "SPG". As it stands, "SPG" means different things for different variables. You are perfectly right.

- P10.L21: I find it hard to see what's happening on seasonal timescales in Fig. 7. This is right, but the message that I wanted to convey with this figure is that there is a large year-to-year variability between ensemble members, complementing Fig. 6: the amount of vertical mixing is uncorrelated between ensemble members (Fig. 6), and that is reflected in the air-sea heat flux (Fig. 7).

- P10.L25: What is meant by "distinguished ... signal"? That the common drift (eq. (2)) shows the same temporal structure in all ensembles.

- P10.L27: I'm confused. Presumably, Fig. 7 is showing net downward heat flux, with negative climatological values indicating that the atmosphere tends to cool the ocean in the SPG. Then, why would a drift towards more negative fluxes (Fig. 7 lower panel) indicate "less heat is extracted from the ocean"? You are right.

- P10.L29: explanation for the low predictability of what, exactly? Ocean and atmosphere temperature.

- Fig. 8: Why don't you actually show the difference from observed climatology from the PD simulation for T2m and SLP, to support the argument that the model drift represents a return to model's own climatology? My aim was to show that the atmospheric drift mainly takes place in the first few years, in contrast to the oceanic drift that goes on for nearly 10 years (Fig. 1). I agree that showing the difference from observed climatology, or the difference between PD and the year 5-10 average of the present runs would better show the return to own climatology.

- Fig. 9: Anomalies from what climatology? I find this an unconvincing plot, because I see a lot of variation with time at all latitudes (and particularly in the 40-60N range). I do not see how this shows that "the drift takes place within the first year". Own climatology. - In all latitude bands the development is from high positive values to predominantly slightly negative values within the first two years. Your are right that this initial drift is smallest in the 40-60N band.

- P14.L8: Again, it's not clear to me that the drift represents a reduction in the amount of heating of the atmosphere (see comment on P10.L27 above) You are right.

- P14.L8-L20: I find the logic difficult to follow, and hence unconvincing, in this discussion. When I look at Fig. 8 (bottom panel), I see a large cooling signal in the vicinity of Grand Banks and throughout the SPG that is almost certainly related to the ocean model drift (ie, the loss of a realistic North Atlantic Current pathway and the overall weakening of the overturning and gyre circulations that transport heat into the SPG). How exactly does the author come to the conclusion that "an ocean signal that far exceeds the internal variability of the model is not able to impact the atmosphere"? The time scales are different. The drift in the ocean takes much longer time than that in the atmosphere.

- P14.L23: "casting doubt" instead "causing doubts" Thank you.

- P14.L25: I don't think this conclusion is obvious at all, particularly since numerous other decadal prediction studies have demonstrated convincing skill in the North Atlantic (Robson et al. 2012, doi:10.1029/2012GL053370; Yeager et al. 2012, doi:10.1175/JCLI-D-11-00595.1; Hermanson et al. 2014, doi:10.1002/2014GL060420; Yeager et al. 2015, doi:10.1002/2015GL065364). Perhaps these results indeed cast doubts on the feasibility of using EC-Earth for predictions, but such a sweeping conclusion is wholly unjustified. I have shown that the mechanism does not work in EC-Earth, and I have pointed out that the predicting skill is also low in other papers (models). Together this suggests that the systematic impact of ocean heat content anomalies on the atmosphere is small.